# Baricitinib and Pulse Steroids Combination Treatment in Hyperinflammatory COVID-19: A Rheumatological Approach in the Intensive Care Unit

**DOI:** 10.3390/ijms25137273

**Published:** 2024-07-02

**Authors:** Francesco Ferro, Gaetano La Rocca, Elena Elefante, Nazzareno Italiano, Michele Moretti, Rosaria Talarico, Erika Pelati, Katia Valentini, Chiara Baldini, Roberto Mozzo, Luigi De Simone, Marta Mosca

**Affiliations:** 1Rheumatology Unit, Department of Clinical and Experimental Medicine, University of Pisa, 56126 Pisa, Italy; 2Anesthesia and Maternal-Infantile Resuscitation Unit, University Hospital of Pisa, 56124 Pisa, Italy

**Keywords:** COVID-19, interstitial lung diseases, baricitinib, pulse steroids, ferritin

## Abstract

Hyperinflammatory Coronavirus disease 2019 (COVID-19) and rapidly-progressive interstitial lung diseases (RP-ILD) secondary to inflammatory myopathies (IIM) present important similarities. These data support the use of anti-rheumatic drugs for the treatment of COVID-19. The aim of this study was to compare the efficacy of combining baricitinib and pulse steroids with the Standard of Care (SoC) for the treatment of critically ill COVID-19 patients. We retrospectively enrolled consecutive patients admitted to the Intensive Care Unit (ICU) with COVID-19-pneumonia. Patients treated with SoC (dexamethasone plus remdesivir) were compared to patients treated with baricitinib plus 6-methylprednisolone pulses (Rheuma-group). We enrolled 246 patients: 104/246 in the SoC and 142/246 in the Rheuma-group. All patients presented laboratory findings suggestive of hyperinflammatory response. Sixty-four patients (26.1%) died during ICU hospitalization. The mortality rate in the Rheuma-group was significantly lower than in the SoC-group (15.5 vs. 40.4%, *p* < 0.001). Compared to the SoC-group, patients in the Rheuma-group presented significantly lower inflammatory biomarker levels after one week of treatment. Higher ferritin levels after one week of treatment were strongly associated with mortality (*p* < 0.001). In this large real-life COVID-19 cohort, baricitinib and pulse steroids led to a significant reduction in mortality, paralleled by a prompt reduction in inflammatory biomarkers. Our experience supports the similarities between hyperinflammatory COVID-19 and the IIM-associated RP-ILD.

## 1. Introduction

Since the beginning of 2020, the worldwide diffusion of severe acute respiratory syndrome coronavirus-2 (SARS-CoV-2) has undermined the very foundations of the entire global health system [1,2]. SARS-CoV-2, an enveloped positive single-stranded RNA virus, is the causative agent for Coronavirus disease 2019 (COVID-19). The 5′ terminal region of the viral genome encodes proteins essential for virus replication, while the 3′ terminal includes five structural proteins: Spike protein (S), membrane protein (M), nucleocapsid protein (N), envelope protein (E), and hemagglutinin–esterase protein (HE) [3].

Indeed, before effective vaccinations and adequate treatment protocols became available, severe cases of COVID-19 exhibited an impressive mortality rate, mainly due to acute respiratory distress syndrome (ARDS) leading to pulmonary and multi-organ failure [4,5]. Notably, due to the ubiquitous distribution of angiotensin-converting enzyme 2 (ACE2), the virus receptor, in different human tissues, COVID-19 may potentially cause numerous extra-pulmonary manifestations. Other life-threatening systemic inflammatory complications may also occur, including disseminated intravascular coagulation and severe hemophagocytosis [6].

At first, research efforts regarding COVID-19 treatment strategies focused on antiviral agents, convalescent plasma, and neutralizing antibodies, while the beneficial effects of glucocorticoid (GC) therapy were questioned due to concerns for delayed viral clearance [7,8]. Nevertheless, growing evidence suggested that severe COVID-19 cases typically show a biphasic natural history. The initial “viral” phase is responsible for mild symptoms and usually lasts about seven days [9]. However, in predisposed subjects, a defective clearance of SARS-CoV-2 in the upper respiratory airways triggers a dysregulated innate and adaptive immune response, leading to inflammatory organ damage. Indeed, severe COVID-19 is a systemic hyperinflammatory (HI) disease characterized by an unbalanced activation of Interferon (IFN) signaling, dysregulation of cellular immune elements, and finally abnormal circulating levels of inflammatory cytokines (cytokine storm), amplifying the inflammatory cascade in a self-perpetuating mechanism [6,10]. This complex pathogenetic process specifically targets lung parenchyma, causing COVID-19 pneumonia, which is characterized by pulmonary consolidations on imaging, and diffuse alveolar damage and microvascular thrombosis in autoptic studies, potentially leading to rapidly progressive respiratory failure, usually in the second week after SARS-CoV-2 infection [11].

As a consequence, it soon became clear that treatment of severe COVID-19 cases should rely on anti-inflammatory and immunomodulatory agents aimed at counteracting the HI process triggered by SARS-CoV-2. The RECOVERY trial first proved the efficacy of daily administration of 6 mg of Dexamethasone for up to 10 days in reducing the 28-day mortality of hospitalized COVID-19 patients requiring respiratory support [12]. Following the preliminary results of this trial, the first WHO guidelines strongly recommended GC therapy for COVID-19 patients with severe and critical disease [13].

At the same time, thanks to the help of artificial intelligence, Stebbing and colleagues identified baricitinib (a JAK 1/2 inhibitor) as a potential effective agent for the treatment of severe COVID-19 [14]. Indeed, in vitro studies showed that baricitinib exerted both a wide-spectrum anti-inflammatory action and a specific antiviral effect against SARS-CoV-2. The former is due to the inhibition of the signaling of multiple pro-inflammatory cytokines, while the latter is accountable to the suppression of clathrin-mediated endocytosis of SARS-CoV-2 virions [15]. Subsequent clinical studies confirmed the efficacy of baricitinib in the treatment of severe COVID-19 patients, however the molecule was introduced in WHO clinical guidelines only in January 2022 [13,16,17,18,19,20].

In March 2020, Italy was the first country to deal with the SARS-CoV-2 outbreak in Europe, and rheumatologists practicing in COVID-hub centers were involved in the first line in the management of hospitalized cases due to their expertise in the use of immunosuppressive drugs. At that time, we pointed out that the pandemics would represent a new challenge for rheumatologists, due to the striking similarities between the pathophysiology and clinical characteristics of severe COVID-19 and other well-known virus-induced autoimmune inflammatory diseases [21,22]. However, whether treatment of this subset of COVID-19 patients with a rheumatological approach would provide significant clinical benefits has not been investigated so far. Therefore, in the present study, we describe our experience in the management of a large Italian cohort of critically ill COVID-19 patients during the period ranging from January 2021 to January 2022. Starting from evidence available in the literature at that time and rheumatological experience in the treatment of autoimmune rapidly-progressive interstitial lung disease (RP-ILD), we managed COVID-19 pneumonia patients admitted to the Intensive Care Unit (ICU) of our hospital with a “rheumatological approach” based on the administration of IV 6-methylprednisolone (6MP) pulses followed by rapid tapering and baricitinib. The aim of this study was to compare the outcome of the patients treated with a “rheumatological approach” to that of critically ill COVID-19 patients treated with a “conventional approach” (dexamethasone plus remdesevir).

## 2. Results

Between 1st September 2020 and 31st January 2022, a total of 246 adult patients hospitalized in the ICU for COVID-19-related pneumonia were consecutively enrolled. Among the patients, 104/246 received the SoC treatment, and 142/246 received the Rheumatologic treatment (Rheuma group).

Patients were mainly male (65%), of Caucasian ethnicity, and presented a mean age of 65.3 (±11) years. Baseline characteristics were similar between the two treatment groups. Namely, no significant differences emerged in demographics, cardiovascular and pulmonary comorbidities (obesity, diabetes, hypertension, COPD), smoking habits, and mean latency from the first COVID-19 related symptom to ICU admission.

At the time of ICU admission, all the enrolled patients exhibited laboratory alterations suggestive of a hyperinflammatory response (C-reactive protein (CRP) 10.8 ± 7.9 mg/dL, ferritin 1224 ± 1018 ng/mL, fibrinogen 573 ± 165 mg/dL), without significant differences between patients in the SoC and the Rheuma-group. The severity of respiratory impairment, in terms of mean PaO_2_/FiO_2_ ratio, was also comparable between the two treatment groups. Baseline characteristics of the study cohort are summarized in Table 1.

Baseline CT scans of all the enrolled patients showed a variable combination of multiple and diffuse areas of increased lung attenuation ranging from ground glass opacities (GGO) to pulmonary consolidations and septal thickening (Figure 1).

A total of 64/246 (26.1%) patients died during ICU hospitalization. Notably, the intra-ICU mortality rate in the Rheuma group was significantly lower compared to that in the SoC group (22/143, 15.5% vs. 42/104, 40.4%; *p* < 0.001). On the other hand, duration of stay in the ICU was not significantly different between the two treatment groups.

After one week of treatment, patients in the Rheuma group showed significantly lower levels of inflammatory biomarkers when compared to the SoC group: CRP 2.0 ± 2.7 vs. 6.1 ± 7.3 mg/dL (*p* < 0.001); fibrinogen 358 ± 118 vs. 453 ± 172 mg/dL (*p* < 0.001); ferritin 888.3 ± 619 vs. 1075 ± 857 ng/mL (*p* < 0.05) (Table 2). Figure 2 shows an example of the evolution of clinical conditions, CRP serum levels, and CT findings of a patient in the Rheuma group.

Within the Rheuma group, no significant correlation was found between baseline TSS-score and serum ferritin values (r = −0.2). Moreover, no significant differences were found between deceased and surviving patients in terms of gender, baseline inflammatory biomarkers, and Total Severity Score (TSS). Importantly, in the Rheuma group, surviving patients presented a shorter time interval from ICU admission to the initiation of treatment (*p* < 0.01) and lower levels of CRP, ferritin, and fibrinogen after one week of treatment (*p* < 0.05). These data are summarized in Table 3.

## 3. Discussion

In the early phases of COVID-19 pandemic, our group and other authors had already hypothesized that the HI phenotype of COVID-19 pneumonia closely resembled RP-ILD developing in the context of inflammatory myopathies (IIM) and might be managed similarly [21,22,23,24]. Particularly, antisynthetase syndrome (ASS)- and anti-MDA5+ dermatomyositis-related ILD share important clinical and pathophysiological similarities with the HI phenotype of COVID-19 pneumonia. Some authors even demonstrated the presence of specific IIM-related autoantibodies, including anti-MDA5 and anti-Ro52, in the sera of hospitalized COVID-19 pneumonia patients, with higher titers associated with a worse outcome [25,26,27,28,29]. In light of this evidence, in the present study we investigated for the first time the efficacy of a “rheumatological approach” based on the combination of baricitinib and pulse high doses of 6MP, for the management of critically ill COVID-19 patients. Our hypothesis was that such a therapeutic regimen aimed at early induction and maintenance of remission could improve the very poor outcome of this subset of patients.

Of note, patients treated with IV 6MP pulses plus baricitinib 4 mg/die for 10–14 days showed a dramatically lower all-cause intra-ICU mortality compared to the SoC treatment group (15.5% vs. 40.4%), despite no significant differences between the two groups in terms of pro-inflammatory biomarkers at baseline, age and other clinical risk factors previously associated with poor prognosis of COVID-19 pneumonia.

Although GC represent the cornerstone of different therapeutic algorithms for severe COVID-19 following the results of the RECOVERY trial, the additional value of GC pulses has long been debated [30,31,32,33]. Notably, a recent double-blind RCT from Salvarani and colleagues failed to demonstrate reduced mortality in hospitalized patients with severe COVID-19 and treated with 6MP pulses [34]. However, unlike in our study, in this trial the most severe patients requiring ICU-admittance were excluded. Actually, we believe that GC pulses may play an important role in selected critically ill COVID-19 cases, especially with an HI phenotype. Indeed, the rationale behind the administration of 6MP pulses lies in the robust and rapid non-genomic anti-inflammatory effects exerted by high-dose GC, which may help to revert the severe autoimmune-mediated ARDS and the cytokine storm characterizing critically ill COVID-19 patients [35]. Finally, the type of GC employed may also play a role, with 6MP pulses exhibiting intermediate duration of action, which allows gradual tapering according to the evolution of the clinical picture. On the opposite side, dexamethasone is a long-acting GC that may expose patients to higher risk of metabolic and infectious complications [36,37].

Regarding baricitinib, several RCTs demonstrated its efficacy and safety in the treatment of hospitalized COVID-19 patients [16,17,38]. Nevertheless, very few studies have evaluated the administration of baricitinib to critically-ill COVID-19 patients. In an RCT enrolling COVID-19 patients requiring invasive mechanical ventilation or extracorporeal membrane oxygenation (ECMO), baricitinib administration resulted in a 19% absolute risk reduction of 28-days mortality [39]. On the other hand, in the subgroup of critically ill patients requiring invasive ventilation or ECMO enrolled in the ACTT2 trial, baricitinib was not able to improve mortality [16]. However, in the ACTT2 trial, only a small proportion of patients received concomitant GC, while literature data suggest that baricitinib exerts a synergistic effect with GC therapy [40,41]. Importantly, our study confirms in a real-life setting the efficacy of baricitinib combined with pulse steroids in severe HI COVID-19.

Interestingly, treatment with the “rheumatological approach” did not result in a shorter duration of stay in the ICU compared to the SoC treatment group. However, this is probably related to the higher short-term mortality in the control group, resulting in a shorter mean hospitalization, which also occurred in previous COVID-19 RCTs [17,39].

In parallel to the reduction of mortality, we registered a greater decline of inflammatory biomarkers after one week of treatment in patients treated with the combination of 6MP pulses and baricitinib, compared to the SoC regimen. This finding is particularly significant considering the key role of systemic hyper-inflammation in determining the progression of COVID-19 pneumonia, with several studies emphasizing the correlation of inflammatory markers with an adverse pulmonary and global outcome [42,43]. Particularly, elevated serum ferritin levels have been widely linked to a severe pulmonary picture and adverse outcome both in COVID-19 pneumonia and IIM-related RP-ILD [44,45,46,47,48,49,50]. Interestingly, while both ferritin and TSS were previously associated with the outcome of COVID-19 pneumonia, baseline values of ferritin and TSS were not significantly correlated in our cohort [51,52]. A possible explanation is that TSS score merely accounts for the extension of pulmonary parenchymal abnormalities on CT, but is not able to capture the severity of pulmonary inflammatory infiltrates. Serum ferritin, in contrast, is likely to better reflect the pulmonary and systemic hyperinflammatory state.

Notably, while early reports suggested male sex was a risk factor for an adverse outcome of COVID-19 pneumonia, we did not observe a difference in male gender prevalence between surviving and deceased patients within the Rheuma group. 

Moreover, our results underline the importance of proper timing in the initiation of immunomodulatory treatment in this subset of patients. Indeed, within the Rheuma group, deceased patients received the combined immunosuppressive treatment later, after ICU-admission, compared to survived patients, and exhibited a smaller improvement of serum inflammatory markers after treatment, suggesting that the systemic HI response was already consolidated and could not be reversed, possibly due to delayed initiation of treatment and/or sub-optimal baricitinib dosage [53]. These findings highlight yet another parallel between HI phenotypes of COVID-19 and IIM-related RP-ILD, which also exhibit a very narrow therapeutic window [54]. Actually, the clinical, serologic and radiological picture of patients included in this study was, overall, very similar to the most widely accepted definitions of RP-ILD in the current IIM literature [55,56]. Besides, numerous studies highlighted pathogenetic pathways shared by the two conditions. Particularly, the hyperexpression of pivotal cytokines including IFN-gamma and IL-6 seems to drive lung inflammatory and fibrotic changes, both in COVID-19 and IIM-related RP-ILD [57,58,59]. It is noteworthy that both IFN-gamma and IL-6 signaling is mediated by Jak1/Jak2 dimerization, leading to STAT3 activation, which is thought to promote several biological processes involved in ILD initiation and progression [60]. The selective Jak1/Jak2 inhibitory action exerted by baricitinib counteracts these pivotal pathogenetic axes, providing the physiopathological rationale behind its success in the management of HI COVID-19 pneumonia [61].

Interestingly, previous studies assessed the efficacy of Tofacitinib, a Jak1/Jak3 inhibitor, for the treatment of ILD associated with anti-MDA5+ IIM, showing promising results [62,63]. However, only three cases of IIM-related ILD treated with baricitinib have been recently reported [64]. In light of the discussed pathogenetic and clinical similarities between COVID-19 pneumonia and RP-ILD, we believe that the selective Jak1/2 inhibitory action of baricitinib may provide additional clinical benefits compared to Tofacitinib.

We recognize that this study presents some limitations. First of all, the retrospective design does not allow us to draw strong conclusions from high quality evidence. Since the two treatment groups compared in the study belong to different phases of the pandemics (September 2020–December 2020, and January 2021–January 2022, respectively), enrolled patients were most likely infected by different variants of SARS-CoV-2, and genotypization was available for a minority of cases. Similarly, data regarding the viral load of enrolled patients were not available. However, the baseline clinical, radiological and laboratory characteristics of critically ill COVID-19 patients were comparable in the two groups, suggesting that in this selected group of patients, differences in the infective SARS-CoV-2 variants have a minor impact on the clinical phenotype. Finally, since the study was conducted during the last months of 2020 and 2021, the SoC treatment included dexamethasone but no other immunomodulatory agents, such as tocilizumab and baricitinib itself, which are nowadays recommended for the treatment of severely and critically ill COVID-19 patients.

On the other hand, the present work has different strengths and important implications, particularly for the rheumatological community. To the best of our knowledge, this is the first study assessing the efficacy of a combined protocol of pulse 6MP and baricitinib for the management of COVID-19 patients in an ICU setting. Moreover, despite the retrospective design, enrolled patients were managed in the same ICU by the same anesthesiology team. Finally, all the enrolled patients were well characterized from a clinical, radiological and laboratory point of view, particularly with repeated assessments of HI biomarkers at definite timepoints.

## 4. Materials and Methods

### 4.1. Participants

This was an observational retrospective monocentric study including adult patients consecutively hospitalized for SARS-CoV-2 respiratory complications in the intensive care unit (ICU) of the Azienda Ospedaliero–UniversitariaPisana (AOUP) (Pisa COVID Hospital), between September 2020 and January 2022. The study was conducted in accordance with the Declaration of Helsinki.

Inclusion criteria were:SARS-CoV-2 infection confirmed by PCR analysis on nasopharyngeal swab;CT-proven SARS-CoV-2 pneumonia;Need for invasive or non-invasive ventilatory support for P/F < 200 at baseline;Absence of clinical and/or laboratory signs of other concomitant infections at the time of admission in ICU (enrollment).

### 4.2. Clinical and Laboratory Parameters

For each patient, the following clinical and laboratory data were collected: demographics, days from first symptom to ER (emergency room) admission, days from first symptom to treatment, duration of ICU-hospitalization, days from ICU-admission to treatment, total duration of hospitalization, laboratory data (C-reactive protein, fibrinogen, ferritin, d-dimer, lactate-dehydrogenase, lymphocytes, procalcitonin) measured before treatment and after one week, cardiopulmonary and vascular comorbidities (COPD, coronary heart disease, diabetes, obesity, hypertension, smoking habits), previous treatment (dexamethasone, remdesivir, macrolides), “rheumatologic” treatment (methylprednisolone pulses, baricitinib, endovenous immunoglobulins) and patient outcome.

### 4.3. CT Scan Description

All CT scans were acquired at the admission in the Emergency Department of AOUP and were evaluated by trained expert radiologists in COVID and non-COVID related interstitial lung disease (ILD). Extension, severity and type of lung lesions were collected. TSS was determined only for patients admitted after January 2021.

### 4.4. Therapeutic Approach and Study Design

All patients received standard of care anesthesiologic treatment, invasive or non-invasive ventilatory support and prophylaxis for venous thromboembolic events to the best of local clinical practice.

Patients enrolled from 1st September 2020 to 31st December 2020 (Standard of Care, “SoC” group), according to literature data available and national and international recommendations of the time, received Standard of Care (SoC) therapy defined as follows: dexamethasone 6 to 8 mg/daily plus remdesivir (200 mg IV. loading dose at day 1, followed by daily 100 mg IV for a total duration of 5–10 days). Some of these patients also received prophylactic antibiotic treatment (mainly beta-lactams or macrolides) and/or hydroxychloroquine (200–400 mg/daily), according to treating physician choice.

Patients enrolled from 1st January 2021 to 31 January 2022 (“Rheuma” group) received a “rheumatologic treatment” defined as follows: 6-methylprednisolone IV pulses (250 or 500 mg daily for three consecutive days followed by rapid tapering) plus baricitinib 4 mg daily for 10–14 days. All patients also received remdesivir (200 mg IV. loading dose at day 1, followed by daily 100 mg IV for a total duration of 5–10 days). Baricitinib dosage was reduced to 2 mg daily in patients with estimated glomerular filtration rate (eGFR) lower than 60 mL/min per 1.73 m^2^. Baricitinib administration was delivered via nasogastric tube after tablet crushing, or orally when feasible.

### 4.5. Study Objectives

The primary objective of the study was to compare intra-ICU mortality between the two groups (SoC group vs. Rheuma group).

The secondary objectives of the study were: (1) to compare inflammatory biomarkers at one week after treatment initiation between the two groups; (2) to compare the duration of stay in ICU due to the need for ventilatory support between the two groups; (3) to analyze clinical and laboratory factors associated with mortality in the Rheuma group.

### 4.6. Statistical Analysis

Data were expressed as mean ± SD for continuous variables, and as absolute frequencies and percentages for categorical variables. Gaussian distribution of variables was assessed using Kolmogorov–Smirnov one-sample test. Chi-Square test and Student’s t-test were performed for comparisons of categorical variables and continuous variables, respectively. Non-parametric tests, such as the Kruskal–Wallis and Mann–Whitney tests, were used for variables that were not normally distributed. SPSS version 29.0.2.0 was used to perform statistical analysis.

## 5. Conclusions

In this real-life study, the combination of 6MP pulses and baricitinib significantly reduced the intra-ICU mortality rate of COVID-19 patients with HI characteristics, supporting the efficacy of a rheumatological approach based on the concepts of phenotyping, combined treatment and window of opportunity, aiming at early induction of remission.

Moreover, our results indirectly confirm the remarkable similarities between the HI phenotype of COVID-19 pneumonia and IIM-related RP-ILD, both in terms of clinical course and treatment response. In conclusion, we believe that the lessons learned from the management of critically ill COVID-19 patients could be translated back to the care of patients with RP-ILD and associated autoimmune systemic diseases, which still present very high mortality rates and several unmet needs.

## Figures and Tables

**Figure 1 ijms-25-07273-f001:**
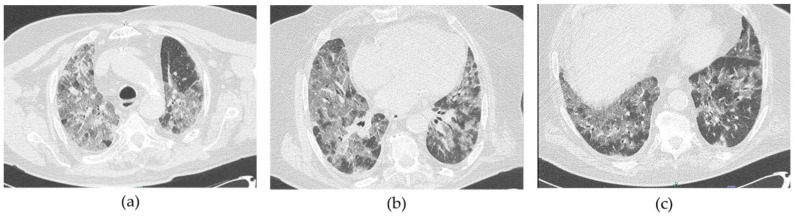
(**a**–**c**). Lung CT-scan sections showing diffuse bilateral and confluent ground-glass opacities with a tendency to lung consolidation, inter- and intra-lobular septal thickening in an enrolled COVID-19 pneumonia patient with severe respiratory failure.

**Figure 2 ijms-25-07273-f002:**
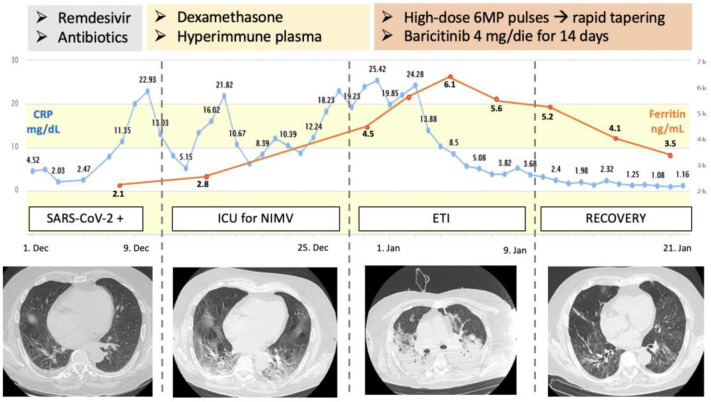
Graphic representation of the C-reactive protein and Ferritin serum levels, clinical course, and corresponding CT findings of the first patient treated with a “Rheumatological approach” after SoC failure. CRP = C Reactive Protein; NIMV = Non-Invasive Mechanical Ventilation; ETI = Endotracheal Intubation.

**Table 1 ijms-25-07273-t001:** Baseline characteristics of the two groups.

	Standard of Care Group (n° = 104)	Rheuma Group (n° = 142)
Age, years (mean ± SD)	66.8 ± 10.4	64 ± 11.3
Sex, n° (%)		
Male	70 (67%)	90 (63%)
Female	34 (33%)	52 (37%)
Pre-existing comorbidities, n° (%)		
COPD ^1^	18 (17%)	29 (20%)
Obesity	11 (10%)	25 (17%)
CHD ^2^	9 (8%)	17 (11%)
Hypertension	25 (24%)	31 (22%)
Inflammatory markers, (mean ± SD)		
C-reactive protein, mg/dL	10.3 ± 8.7	11.6 ± 6.6
Ferritin, ng/mL	1155 ± 1059	1294 ± 995
Fibrinogen, mg/dL	568 ± 200	580 ± 130
Lactate dehydrogenase, U/L	395 ± 165	382 ± 130
Lymphocytes/mcL	810 ± 240	480 ± 99
Max d-dimer, mg/L	1410 ± 749	3507 ± 3044
P/F ^3^ Ratio, (mean ± SD)	203 ± 122	211 ± 143
Time from first symptom to ICU ^4^, days (mean ± SD)	15.0 ± 4.2	9.5 ± 4.5

^1^ Chronic obstructive pulmonary disease. ^2^ Coronary heart disease. ^3^ PaO_2_/FiO_2_ ratio. ^4^ Intensive care unit. No significant differences were observed between SoC and Rheuma group for all the variables in this table (*p* > 0.05).

**Table 2 ijms-25-07273-t002:** Outcome comparison between the two treatment groups.

	Standard of Care Group (n° = 104)	Rheuma Group (n° = 142)	*p* Value
Intra ICU ^1^ death rate, n° (%)	42/104 (40.4%)	22/142 (15.5%)	<0.001
Duration of ICU ^1^ hospitalization, days (mean ± SD)	14.6 ± 9.6	14.5 ± 8.1	ns
Inflammatory markers after 1 week (mean ± SD)			
C-reactive protein, mg/dL	6.14 ± 7.3	2.03 ± 2.75	<0.001
Ferritin, ng/mL	1075 ± 856	888 ± 619	<0.05
Fibrinogen, mg/dL	453 ± 172	358 ± 118	<0.001

^1^ Intensive care unit. ns = not statistically significant.

**Table 3 ijms-25-07273-t003:** Intra-Rheuma Group comparison between survived and deceased patients.

	Survivors (n° = 104, 84%)	Deceased (n° = 22, 16%)	*p* Value
Age, years (mean ± SD)	62.9 ± 11.5	70.9 ± 7.1	<0.01
Sex, n° (%)			
Male	76 (63%)	14 (63%)	ns
Female	28 (27%)	8 (27%)	ns
Inflammatory markers at baseline (mean ± SD)			
C-reactive protein, mg/dL	6.27 ± 2.55	15.67 ± 7.59	ns
Ferritin, ng/mL	744 ± 409	737 ± 205	ns
Fibrinogen, mg/dL	590 ± 119	577 ± 132	ns
Lactate dehydrogenase, U/L	375 ± 44	420 ± 58	ns
Procalcitonin ng/mL	0.09 ± 0.04	0.13 ± 0.02	ns
Baseline TSS ^1^	11.8 ± 3.2	12.6 ± 4.3	ns
Inflammatory markers after one week (mean ± SD)			
C-reactive protein, mg/dL	1.47 ± 1.76	5.31 ± 4.60	<0.001
Ferritin, ng/mL	842 ± 554	1140 ± 871	<0.05
Fibrinogen, mg/dL	344 ± 100	441 ± 169	<0.001
Lactate dehydrogenase, U/L	375 ± 44	420 ± 58	ns
Duration of symptoms before hospitalization, days (mean ± SD)	8.2 ± 3.1	6.5 ± 2.5	ns
Time between hospitalization and treatment start, days (mean ± SD)	5.1 ± 4.2	8.2 ± 6.8	<0.01
Total duration of ICU ^2^ hospitalization, days (mean ± SD)	13.5 ± 3.5	17.3 ± 8.4	ns

^1^ Total severity score. ^2^ Intensive care unit. ns = not statistically significant.

## Data Availability

Data are available upon reasonable request.

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
