# Peer review of "Baricitinib and Pulse Steroids Combination Treatment in Hyperinflammatory COVID-19: A Rheumatological Approach in the Intensive Care Unit"

_ijms, 2024, doi:10.3390/ijms25137273_

Round 1
Reviewer 1 Report
Comments and Suggestions for Authors
This manuscript compares the outcome of the patients treated with a “rheumatological approach” to that of critically-ill COVID-19 patients treated with a “conventional approach”(dexamethasone plus Remdesevir).
This manuscript shows rich content, providing a deep insight for some works: the study is within the journal’s scope, and I found it to be well-written, providing sufficient information. Even if the manuscript provides an organic overview, with a densely organized structure and based on well-synthetized evidence, there are some suggestions necessary to make the article complete and fully readable. For these reasons, the manuscript requires major changes.
Please find below an enumerated list of comments on my review of the manuscript:
MINOR POINTS:
The authors should provide a list of the abbreviations, mentioned in this manuscript.
MAJOR POINTS:
ABSTRACT:
LINE 19: Please, rephrase this sentence as following: “The aim of this manuscript was to compare the efficacy of combining Baricitinib and pulse steroids with the Standard of Care (SoC) for the treatment of critically-ill COVID-19 patients.
INTRODUCTION:
LINE 38: Since the beginning of 2020, the worldwide diffusion of severe acute respiratory syndrome coronavirus-2 (SARS-CoV-2) undermined the very foundations of the entire global health system.
LINE 39: The causative agent for COVID-19, SARS-CoV-2, is an enveloped positive single-stranded RNA virus, whose viral genome includes a 5’ terminal which encodes proteins essential for virus replication, and the 3’ terminal includes five structural proteins, Spike protein (S), membrane protein (M), nucleocapsid protein (N), envelope protein (E), and hemagglutinin-esterase protein (HE) (see, for reference: Lu, R.; Zhao, X.; Li, J.; Niu, P.; Yang, B.; Wu, H.; Wang, W.; Song, H.; Huang, B.; Zhu, N.; et al. Genomic characterisation and epidemiology of 2019 novel coronavirus: Implications for virus origins and receptor binding. Lancet 2020, 395, 565–574).
LINE 42: As a global health threat, Coronavirus disease 2019 (COVID-19) was characterized by significant pulmonary and extrapulmonary involvement: in fact, due to a ubiquitous distribution of ACE-2 in different organs, SARS-CoV-2 infection may affect the lungs primarily, leading to respiratory failure; however, this infection simultaneously involves several organs, from kidneys to the heart, blood vessels, liver, pancreas, and immune system. According to recent scientific evidence on this topic, this manuscript may benefit from highlighting the importance of extra-pulmonary manifestations, associated to COVID-19. This is the major concern of this manuscript, whose clarification will improve the impact and quality of this manuscript.
METHODS:
LINE 328: If possible, the authors should specify the software applied to perform the statistical analysis.
This is an interesting manuscript, whose originality is due to the extension of the research field on the outcome of the patients treated with a “rheumatological approach” to that of critically-ill COVID-19 patients treated with a “conventional approach”(dexamethasone plus Remdesevir), as a further extra-pulmonary manifestation of COVID-19. The main topic is of great clinical impact. At the same time, the contents are rich, with a deep insight for some works.
Furthermore, there is a specific and detailed explanation for the methods used in this study: this is particularly significant, since the manuscript relies on a multitude of methodological and statistical analysis, to derive its conclusions. The methodology applied is overall correct, the results are reliable and adequately discussed.
The conclusion of this manuscript is perfectly in line with the main purpose of the paper: the authors have designed and conducted the study properly. As regards the conclusions, they are well written and present an adequate balance between the description of previous findings and the results presented by the authors.
In conclusion, this manuscript is well organized, based on well-synthetized evidence. Besides, the methodology design was appropriately implemented within the study. However, many of the topics are very concisely covered. This manuscript provided a comprehensive analysis of current knowledge in this field. However, major concerns of this manuscript are with the introductive section: for these reasons, I have major comments for this section, for improvement before acceptance for publication. I have some major points to make, that may help to improve the quality of the current manuscript and maximize its scientific impact.
Author Response
Dear reviewer, thank you for the time you dedicated to evaluate our manuscript and for your valuable feedback. We completely accepted all your suggestions and revised the manuscript accordingly as detailed below.
“This manuscript shows rich content, providing a deep insight for some works: the study is within the journal’s scope, and I found it to be well-written, providing sufficient information. Even if the manuscript provides an organic overview, with a densely organized structure and based on well-synthetized evidence, there are some suggestions necessary to make the article complete and fully readable. For these reasons, the manuscript requires major changes.
Please find below an enumerated list of comments on my review of the manuscript:
MINOR POINTS:
The authors should provide a list of the abbreviations, mentioned in this manuscript.
Thank you for the suggestion, we added a list of abbreviations
MAJOR POINTS:
ABSTRACT:
LINE 19: Please, rephrase this sentence as following: “The aim of this manuscript was to compare the efficacy of combining Baricitinib and pulse steroids with the Standard of Care (SoC) for the treatment of critically-ill COVID-19 patients.
Thank you for the suggestion, we rephrased the sentence.
INTRODUCTION:
LINE 38: Since the beginning of 2020, the worldwide diffusion of severe acute respiratory syndrome coronavirus-2 (SARS-CoV-2) undermined the very foundations of the entire global health system.
LINE 39: The causative agent for COVID-19, SARS-CoV-2, is an enveloped positive single-stranded RNA virus, whose viral genome includes a 5’ terminal which encodes proteins essential for virus replication, and the 3’ terminal includes five structural proteins, Spike protein (S), membrane protein (M), nucleocapsid protein (N), envelope protein (E), and hemagglutinin-esterase protein (HE) (see, for reference: Lu, R.; Zhao, X.; Li, J.; Niu, P.; Yang, B.; Wu, H.; Wang, W.; Song, H.; Huang, B.; Zhu, N.; et al. Genomic characterisation and epidemiology of 2019 novel coronavirus: Implications for virus origins and receptor binding. Lancet 2020, 395, 565–574).
We implemented the introduction with more detailed information regarding SARS-CoV-2
LINE 42: As a global health threat, Coronavirus disease 2019 (COVID-19) was characterized by significant pulmonary and extrapulmonary involvement: in fact, due to a ubiquitous distribution of ACE-2 in different organs, SARS-CoV-2 infection may affect the lungs primarily, leading to respiratory failure; however, this infection simultaneously involves several organs, from kidneys to the heart, blood vessels, liver, pancreas, and immune system. According to recent scientific evidence on this topic, this manuscript may benefit from highlighting the importance of extra-pulmonary manifestations, associated to COVID-19. This is the major concern of this manuscript, whose clarification will improve the impact and quality of this manuscript.
We acknowledged the potential for COVID-19 to cause numerous extrapulmonary manifestations due to the diffuse distribution of ACE2 in human tissues.
METHODS:
LINE 328: If possible, the authors should specify the software applied to perform the statistical analysis.
We specified in the methods section the employed software.
This is an interesting manuscript, whose originality is due to the extension of the research field on the outcome of the patients treated with a “rheumatological approach” to that of critically-ill COVID-19 patients treated with a “conventional approach”(dexamethasone plus Remdesevir), as a further extra-pulmonary manifestation of COVID-19. The main topic is of great clinical impact. At the same time, the contents are rich, with a deep insight for some works. Furthermore, there is a specific and detailed explanation for the methods used in this study: this is particularly significant, since the manuscript relies on a multitude of methodological and statistical analysis, to derive its conclusions. The methodology applied is overall correct, the results are reliable and adequately discussed. The conclusion of this manuscript is perfectly in line with the main purpose of the paper: the authors have designed and conducted the study properly. As regards the conclusions, they are well written and present an adequate balance between the description of previous findings and the results presented by the authors. In conclusion, this manuscript is well organized, based on well-synthetized evidence. Besides, the methodology design was appropriately implemented within the study. However, many of the topics are very concisely covered. This manuscript provided a comprehensive analysis of current knowledge in this field. However, major concerns of this manuscript are with the introductive section: for these reasons, I have major comments for this section, for improvement before acceptance for publication. I have some major points to make, that may help to improve the quality of the current manuscript and maximize its scientific impact."
Thank you very much for your appreciation and mostly for your valuable suggestions regarding the introduction of the manuscript.
We believe the revised version was improved by your suggestion and we hope it is now acceptable for publication.
Best regards on behalf of all the authors
Reviewer 2 Report
Comments and Suggestions for Authors
In this interesting study, authors compare two kinds of treatment regimens for patients with Covid - 19 disease, hospitalized in an Intensive Care Unit.
They compare patients treated with the standard of care with patients treated with a «rheumatologic» regimen based upon the characteristics of the disease and combining 6-methylprednisolone and baricitinib.
They compared two groups in different time points treated with different regimens. They concluded that the treatment including baricitinib was more efficient in terms of mortality rate compared to the standard of care.
The abstract is presented in a well-structured manner.
The introduction is well written and clear.
The manuscript is scientifically sound, and the methods are well written.
The references are recent.
I think that the reference: Acute Crit Care > Volume 38(1); 2023
Infection Methylprednisolone pulse therapy for critically ill patients with COVID-19: a cohort study
Keum-Ju Choi
, Soo Kyun Jung
, Kyung Chan Kim
, Eun Jin Kim
should be included.
Finally the TSS score should be analyzed
Author Response
Dear reviewer, thank you for the time you dedicated to evaluate our manuscript and for your valuable feedback. We completely accepted all your suggestions and revised the manuscript accordingly as detailed below.
In this interesting study, authors compare two kinds of treatment regimens for patients with Covid - 19 disease, hospitalized in an Intensive Care Unit.
They compare patients treated with the standard of care with patients treated with a «rheumatologic» regimen based upon the characteristics of the disease and combining 6-methylprednisolone and baricitinib.
They compared two groups in different time points treated with different regimens. They concluded that the treatment including baricitinib was more efficient in terms of mortality rate compared to the standard of care.
The abstract is presented in a well-structured manner.
The introduction is well written and clear.
The manuscript is scientifically sound, and the methods are well written.
The references are recent.
I think that the reference: Acute Crit Care > Volume 38(1); 2023
Infection Methylprednisolone pulse therapy for critically ill patients with COVID-19: a cohort study
Keum-Ju Choi, Soo Kyun Jung, Kyung Chan Kim, Eun Jin Kim
should be included.
Thank you for your appreciation and for signaling this paper that definitely deserved to be cited.
Finally the TSS score should be analyzed
Thank you for your suggestion, we included a paragraph in the discussion section to analyze findings regarding TSS
We believe the revised version was improved by your suggestion and we hope it is now acceptable for publication.
Best regards on behalf of all the authors
Reviewer 3 Report
Comments and Suggestions for Authors
In the manuscript entitled "Baricitinib and pulse steroids combination treatment in hyper- inflammatory COVID-19: a rheumatological approach in intensive care unit", the authors aimed to analyze the efficacy of combining baricitinib and pulse steroids with the Standard of Care (SoC) in patients with severe COVID-19. The authors confirmed already published results that baricitinib treatment improves survival of patients with severe COVID-19 accompanied by reduction of inflammatory biomarkers. The novelty is not extremely high and is mainly based on the combination of baricitinib with steroids. Please improve language quality to ease readability.
Major concerns.
1) Can the author provide further information about the virus strain and virus load in both patient groups?
2) Since male and female differ in the metabolic activity, the authors should include further information about plasma levels of each drug that might result in different plasma levels in male and females.
3) In figure 1, the authors showed Lung CT-scan sections of an enrolled COVID-19 pneumonia patient with severe respiratory failure, however I am wondering if the authors have data for patients under baricitinib and pulse steroid therapy.
4) In the discussion section, the authors referred to increased presence of specific IIM-related autoantibodies, including anti-MDA5and anti-Ro52, in the sera of hospitalized COVID-19 pneumonia patients. As the authors analyzed the benefical effect of immune modulating drugs have they measured autoantibody concentrations in the sera of patients? Please add more clinical data.
5) Particular with regards to earlier reports that showed a sex bias in the susceptibility and outcome of respiratory virus infection (influenza and SARS-CoV-2), is there a sex difference in the outcome of infection upon combination therapy with baricitinib and steroids?
6) The authors referred to the students t test, is the group size big enough to estimate Gaussian normal distribution, if not the authors should analyze the data by Mann-Whitney-U-test.
7) The authors should expand the discussion section and discuss the possibility of drug combinations. There are several reports of drug combination particular with immune modulating drugs such as baricitinib (PMID: 35677732, PMID: 36410777, PMID: 34902115, PMID: 37927607), p38 inhibitors (PMID: 36423831), or Erk inhibitors (PMID: 36145524). Please elaborate in the discussion section advantages of drug combinations by using the mentioned references.
I hope that the authors can revise the manuscript accordingly to the suggestions, which will definitely increase the scientific soundness and overall merit of the manuscript.
Comments on the Quality of English LanguagePlease improve language quality.
Author Response
Dear reviewer, thank you for the time you dedicated to evaluate our manuscript and for your valuable feedback. We tried our best to implement the manuscript according to your suggestions, as detailed below.
In the manuscript entitled "Baricitinib and pulse steroids combination treatment in hyper- inflammatory COVID-19: a rheumatological approach in intensive care unit", the authors aimed to analyze the efficacy of combining baricitinib and pulse steroids with the Standard of Care (SoC) in patients with severe COVID-19. The authors confirmed already published results that baricitinib treatment improves survival of patients with severe COVID-19 accompanied by reduction of inflammatory biomarkers. The novelty is not extremely high and is mainly based on the combination of baricitinib with steroids. Please improve language quality to ease readability.
Thank you for your comment, language quality was reasonably improved and minor tipos fixed.
Major concerns.
1) Can the author provide further information about the virus strain and virus load in both patient groups?
As we specified in the paragraph acknowledging the study limits in the discussion section, information regarding genotypization was not available. We addedd a sentence to acknowledge the lack of data regarding viral load.
2) Since male and female differ in the metabolic activity, the authors should include further information about plasma levels of each drug that might result in different plasma levels in male and females.
We remind you that this was a retrospective study based on a real life experience in a ICU-setting during the early phases of pandemics in Italy. This was not a clinical trial and plasma levels of drugs administered to critically-ill COVID-19 patients were not routinely assessed in our hospital. Therefore unfortunately these data are not available.
3) In figure 1, the authors showed Lung CT-scan sections of an enrolled COVID-19 pneumonia patient with severe respiratory failure, however I am wondering if the authors have data for patients under baricitinib and pulse steroid therapy.
Thank you for the great suggestion. We included an additional figure representing the evolution of CT findings, clinical conditions and laboratory parameters of a patient treaterd with baricitinib and pulse steroid therapy.
4) In the discussion section, the authors referred to increased presence of specific IIM-related autoantibodies, including anti-MDA5and anti-Ro52, in the sera of hospitalized COVID-19 pneumonia patients. As the authors analyzed the benefical effect of immune modulating drugs have they measured autoantibody concentrations in the sera of patients? Please add more clinical data.
Unfortunately in the clinical setting of this study IIM-related autoantibodies testing was not routinely performed.
5) Particular with regards to earlier reports that showed a sex bias in the susceptibility and outcome of respiratory virus infection (influenza and SARS-CoV-2), is there a sex difference in the outcome of infection upon combination therapy with baricitinib and steroids?
Thank you for your question. As reported in the results section and in table 3, deceased and survived patients treated with the combination of baricitinib and pulse steroids did not significantly differ in terms of gender prevalence. We addedd a period in the discussion session in order to underline this aspect.
6) The authors referred to the students t test, is the group size big enough to estimate Gaussian normal distribution, if not the authors should analyze the data by Mann-Whitney-U-test.
Thank you for noticing this inaccuracy in the methods section. Gaussian distribution of variables was assessed using Kolmogorov-Smirnov one-sample test. than Mann-Whitney-U-test was used for variables not following a normal distribution. Non-parametric tests, such as Kruskal-Wallis and Mann-Whitney tests, were used for variables that were not normally distributed. We corrected the method section (lines 444-449), re-checked the statistical analyses and we adjusted small inaccuracies.
7) The authors should expand the discussion section and discuss the possibility of drug combinations. There are several reports of drug combination particular with immune modulating drugs such as baricitinib (PMID: 35677732, PMID: 36410777, PMID: 34902115, PMID: 37927607), p38 inhibitors (PMID: 36423831), or Erk inhibitors (PMID: 36145524). Please elaborate in the discussion section advantages of drug combinations by using the mentioned references.
Thank you very much for your suggestion. We implemented the manuscript by adding some of the references you recommanded that were functional to the introduction and/or discussion sections, including PMID: 35677732 (Baricitinib in hospitalised patients with COVID-19: A meta-analysis of randomised controlled trials) and PMID: 34902115 (Baricitinib combination therapy: a narrative review of repurposed Janus kinase inhibitor against severe SARS-CoV-2 infection). On the other hand, we feel that in vitro studies regarding combination of anti-viral and new immunomodulatory molecules (PMID: 36423831 and 36145524) fall well beyond the scopes of the discussion of this study. Similarly we did not include PMID: 37927607 (The effects of combination-therapy of tocilizumab and baricitinib on the management of severe COVID-19 cases: a randomized open-label clinical trial) because based on clinical experience and the pathophysiological mechanisms of the two drugs we do not believe that combining baricitinib with anti-IL6R would provide significant advantages in terms of clinical efficacy and safety profile (as showed by the results of the mentioned study).
I hope that the authors can revise the manuscript accordingly to the suggestions, which will definitely increase the scientific soundness and overall merit of the manuscript.
Comments on the Quality of English Language
Please improve language quality.
Thank you for the suggestion. Language quality was reasonably improved and minor typos were fixed.
We believe the revised version was improved by your suggestion and we hope it is now acceptable for publication.
Best regards on behalf of all the authors
Reviewer 4 Report
Comments and Suggestions for Authors
Thanks for the opportunity to review the manuscript entitled ”Baricitinib and pulse steroids combination treatment in hyperinflammatory COVID-19: a rheumatological approach in intensive care unit”.
I enjoyed reading this manuscript.
The discussions and conclusions are useful for further clinical practice and management.
Author Response
Dear reviewer, thank you for the time you dedicated to evaluate our manuscript and for your valuable feedback.
Thanks for the opportunity to review the manuscript entitled ”Baricitinib and pulse steroids combination treatment in hyperinflammatory COVID-19: a rheumatological approach in intensive care unit”.
I enjoyed reading this manuscript.
The discussions and conclusions are useful for further clinical practice and management.
Thank you very much for your appreciation of the manuscript.
Best regards on behalf of all the authors
Round 2
Reviewer 1 Report
Comments and Suggestions for Authors
The authors have significantly improved the scientific impact of this manuscript.